# Co-Extraction of Aluminum and Silicon and Kinetics Analysis in Carbochlorination Process of Low-Grade Bauxite

**DOI:** 10.3390/ma17143613

**Published:** 2024-07-22

**Authors:** Xinxin Zhao, Yan Liu, Long Wang, Yutong Hua, Tianhao Cheng, Tingan Zhang, Qiuyue Zhao

**Affiliations:** 1Key Laboratory for Ecological Utilization of Multimetallic Mineral, Ministry of Education, Northeastern University, Shenyang 110819, China; 2Chinalco Southeast Material Institute (Fujian) Technology Co., Ltd., Fuzhou 350015, China

**Keywords:** low-grade bauxite, carbochlorination, extraction of aluminum and silicon, kinetic study, resource utilization

## Abstract

Addressing the issue that the Bayer process is not suitable for low-grade bauxite, carbochlorination was proposed to recover aluminum and silicon from low-grade bauxite. This study focused on the behavior of aluminum and silicon during the carbochlorination process of low-grade bauxite. The impact of various process parameters on the chlorination efficiency was investigated, and the chlorination mechanism and kinetics of aluminum and silicon chlorination in bauxite were analyzed and discussed. Under optimal experimental conditions, the chlorination efficiency of Al_2_O_3_ and SiO_2_ reached 94.93% and 86.32%, respectively. The carbochlorination of aluminum and silicon in bauxite adhered to a shrinking, unreacted core model governed by gas diffusion within the product layer. This process can be bifurcated into two stages. Additionally, calculations were conducted to determine the apparent activation energy and reaction order of the chlorination processes involving Al_2_O_3_ and SiO_2_. Examining the chlorination mechanism revealed that the bauxite carbochlorination encompasses transformations among various minerals. Notably, the aluminum component prefers to participate in the carbothermal chlorination reaction over silicon.

## 1. Introduction

Bauxite is the primary raw material for the alumina production industry, accounting for over 95% of global alumina production [1]. The Bayer process, traditionally employed for approximately 90% of bauxite treatment [2], is unsuitable for low-grade bauxite with A/S < 5 (mass ratio of alumina to silica). This inadequacy arises from high alkali consumption, substantial red mud production, and a low alumina extraction rate. Given its status as the world’s predominant producer and consumer of aluminum, China has an immense requirement for alumina. Over 95% of China’s bauxite resources are diaspore-type bauxite, with middle- and low-grade bauxite (A/S < 7) constituting more than 70% of total reserves. Due to the scarcity of high-quality bauxite resources, China’s dependence on bauxite imports has exceeded 50% [3]. China’s bauxite reserves are abundant in silicon and iron, and contain titanium, gallium, germanium, and scandium, among other critical metal elements, all of which hold considerable economic value. Therefore, according to the unique characteristics of bauxite in China, it has emerged as a strategic concern for advancing China’s alumina industry to explore novel economical alumina production processes.

Various pre-desilication processes have been proposed to fulfill the A/S ratio requirements of bauxite, including flotation desilication, chemical desilication, and biological desilication [4,5]. These methods aim to reduce the silicon-containing minerals present in bauxite. Flotation desilication is the predominant method utilized in industrial production due to constraints related to production cost or process technology. However, challenges persist in reducing ore grade and recovery and increasing energy consumption during the subsequent dissolution process. The sintering method is suitable for treating bauxite with an A/S ratio of 3~6. Despite this, it has the disadvantages of a relatively complex process, high energy consumption, and lower product quality compared to the Bayer method. The combined method, which integrates the benefits of the Bayer and sintering processes, is optimal for treating middle- to low-grade bauxite with an A/S ratio of 7~9. However, it is unavoidable that this process results in a complex flow and mutual constraints between the two systems.

Chlorination metallurgy is a prevalent technique utilized in the metallurgical industry to treat intricate metal minerals [6]. In 1973, the American Aluminum Company (Alcoa) pioneered a technological method to convert Bayer alumina into aluminum chloride, which was subsequently electrolyzed to produce primary aluminum [7]. This method aimed to reduce electricity consumption by 30% and diminish the labor intensity associated with aluminum refining, and pilot-scale tests were undertaken in 1976 [8]. Nonetheless, the process necessitated additional processing stages in raw material production, and the corrosive impact of recycled chlorine escalated the cost of producing anhydrous aluminum chloride [8]. In 1978, the carbochlorination process of kaolinite and kaolinite-type clay was initially investigated using a fluidized bed reactor, enabling the conversion of Al_2_O_3_ and SiO_2_ to AlCl_3_ and SiCl_4_ [9]. To enhance the production of AlCl_3_ during the carbochlorination process of bauxite or clay, selective inhibition of the chlorination reaction with silicon was achieved by adding SiCl_4_ [10].

Leveraging the characteristics of chlorination metallurgy, the Institute of Special Metallurgy and Process Engineering at Northeastern University proposes a novel method of “carbochlorination–oxygen pressure conversion” to produce alumina from low-grade bauxite. Initially, bauxite, coking coal, and binder mixtures are granulated to fabricate bauxite pellets. These pellets undergo high-temperature carbochlorination, facilitating the conversion of various metal oxides into their corresponding metal chlorides. A preliminary condensation separation is then executed, leveraging differences in melting points and saturation vapor pressures. The purified aluminum chloride, derived from this separation, is subsequently transformed into alumina products under an oxygen pressure environment, while the chlorine gas is recycled for reuse. The chlorination–oxygen pressure conversion method offers several advantages over traditional low-grade bauxite comprehensive utilization. (1) This process transcends the traditional constraints on the A/S ratio of bauxite, offering a broad spectrum of applications. (2) The procedure achieves zero emissions, thereby circumventing secondary pollution induced by waste acid, waste alkali, wastewater, and waste residue. (3) The process flow is concise, energy consumption is minimal, and continuous production can be easily achieved. (4) Chlorine gas can be recycled, thereby reducing production costs. (5) This process facilitates the resource recovery and utilization of all components of bauxite, effectively recovering critical metal resources with significant economic value.

This paper investigated the carbochlorination behavior of aluminum and silicon during the carbochlorination process of typical low-grade bauxite in China. The feasibility of bauxite carbochlorination was assessed through thermodynamic analysis. The study examined the influence of various process parameters, including chlorination time, coke addition amount, chlorination temperature, gas flow rate, and oxygen content, on the chlorination efficiency of the low-grade bauxite. Reaction kinetics analysis identified the vital kinetic parameters for aluminum and silicon during the carbochlorination of low-grade bauxite. Further analysis was carried out to elucidate the carbochlorination mechanism of low-grade bauxite. This research offers novel methods and theoretical insights for enhanced value extraction from low-grade bauxite resources.

## 2. Materials and Methods

### 2.1. Materials

The low-grade bauxite utilized in this study was sourced from an enterprise located in Shanxi Province, China. The coking coal, possessing a fixed carbon content of 65.79%, was procured from a company in Shanxi. The pellet binder utilized in this study was modified starch sourced from Henan Jianjie Industrial Co., Ltd., located in Zhengzhou, China. Cl_2_ (99 vol%) was supplied by Xinfa Chemical Co., Ltd., located in Xiaoyi, China.

The chemical compositions were analyzed using X-ray fluorescence (XRF), as detailed in Table 1. The bauxite primarily comprised Al_2_O_3_ (61.11%) and SiO_2_ (22.49%), making it a typical low-grade bauxite with an A/S ratio of 2.72. As illustrated in Figure 1, the mineralogical analysis reveals that the bauxite predominantly consisted of 61.11% diaspore (AlOOH), 24.24% kaolinite (Al_2_(Si_2_O_5_)(OH)_4_), 5.10% quartz (SiO_2_), 3.42% corundum (Al_2_O_3_), and 1.38% hematite (Fe_2_O_3_).

### 2.2. Methods

Bauxite was combined with coking coal and a modified starch binder in an optimal mole ratio of O to C in bauxite and coking coal, respectively, incorporating 10 wt.% of the modified starch binder. Subsequently, a small quantity of water was sprayed into the mixture to produce pellets with diameters ranging from 8 to 12 mm in the round pot granulator. These pellets were then subjected to a drying process. A quantity of 100 g of bauxite pellets was heated to a specified temperature within a vertical quartz fixed-bed reactor at 20 °C/min, followed by roasting for a designated duration in a mixed atmosphere of chlorine and oxygen. Upon completion of the reaction, the furnace was allowed to cool to room temperature, after which the chlorinated residue was collected. This residue was subsequently washed and dried for subsequent chemical composition analysis and phase characterization.

The contents of Al_2_O_3_ and SiO_2_ in bauxite and chlorinated residue after leaching were determined using an X-ray fluorescence spectrometer (XRF, ZSX PrimusIV, Tokyo, Japan). The chlorination efficiency of Al_2_O_3_ or SiO_2_ is calculated as
(1)$$X=\frac{M\times W-m\times w}{M\times W}\times 100\%$$
where *X* represents the chlorination efficiency expressed as a percentage (%), *M* (g) denotes the initial mass of bauxite, *W* (%) signifies the contents of Al_2_O_3_ or SiO_2_ within the bauxite, *m* (g) indicates the mass of the resultant chlorinated residue, and *w* (%) stands for the contents of Al_2_O_3_ or SiO_2_ in the chlorinated residue.

Furthermore, the phase compositions of bauxite and chlorinated residues were obtained by X-ray diffraction (XRD, Bruker D8 Advance, Berlin, Germany), and data analysis was performed using X’Pert HighScore Plus.

## 3. Results and Discussion

### 3.1. Thermodynamics of Carbochlorination Reactions

The phase analysis of low-grade bauxite, as depicted in Figure 1, reveals that the primary phases containing aluminum and silicon are diaspore, kaolinite, quartz, and corundom. Diaspore and kaolinite undergo a dehydration reaction as the temperature fluctuates, transforming into α-Al_2_O_3_ and metakaolinite (Al_2_O_3_·2SiO_2_), respectively [11]. This transformation is represented by Equations (2) and (3).
(2)AlOOH →500 C∘ α-Al2O3+H2O
(3)Al2O3⋅2SiO2⋅2H2O →500 ~ 600 C∘ Al2O3⋅2SiO2+2H2O

The process of carbochlorination of bauxite, a high-temperature reaction, involves the substitution of α-Al_2_O_3_ and Al_2_O_3_·2SiO_2_ for the original diaspore, with kaolinite participating in the carbochlorination reaction.

The feasibility of extracting aluminum and silicon from bauxite through carbochlorination was evaluated utilizing Gibbs free energy minimization theory [12]. Specifically, the Gibbs free energy (∆*G*^θ^) changes for possible chemical reactions involving aluminum and silicon were calculated using Factsage 8.1 software (FactPS database); considering that 1 mol of Cl_2_ was involved in the reaction, the results are presented in Figure 2. As depicted in Figure 2a, Al_2_O_3_ and SiO_2_ undergo spontaneous carbochlorination reactions within the temperature range of 400~1400 °C. Notably, the ∆*G*^θ^ value for the carbochlorination of Al_2_O_3_ is more damaging than that of SiO_2_ above 700 °C, suggesting a greater ease of occurrence for the chlorination reaction of Al_2_O_3_. Additionally, due to the Boudouard reaction [13], carbochlorination reactions predominantly yield CO rather than CO_2_. The comparison of ∆*G*^θ^ values for the carbochlorination reaction of Al_2_O_3_·2SiO_2_, as depicted in Figure 2b, reveals that at a temperature of 1000 °C, the ∆*G*^θ^ associated with the conversion of Al in Al_2_O_3_·2SiO_2_ to AlCl_3_ is less than that required for the transformation of Si to SiCl_4_. This observation suggests a preferential formation of AlCl_3_ over SiCl_4_ during the chlorination reaction of Al_2_O_3_·2SiO_2_.

Figure 3 presents the predominance diagram for the Al-Si-O-Cl system at temperatures of 800 °C and 1000 °C. The Gibbs free energy minimization principle dictates that all substances depicted in the predominance diagram exist in their most stable state [14]. The findings reveal that a higher partial pressure of chlorine and a lower partial pressure of oxygen favor the formation of AlCl_3_ and SiCl_4_. At a constant oxygen partial pressure, the required partial pressure of chlorine gas for AlCl_3_ formation is less than that for SiCl_4_, suggesting that AlCl_3_ forms preferentially over SiCl_4_. A green zone, comprising gaseous phase AlCl_3_ and SiCl_4_, is present in the predominance diagram at varying temperatures, as shown in Figure 3a,b. This region shifts to the right with increasing temperature, indicating that excessive heat increases the complexity of carbochlorination.

Consequently, extracting aluminum and silicon from low-grade bauxite is thermodynamically feasible. An appropriate augmentation of carbon content can diminish oxygen potential, facilitating the carbochlorination process.

### 3.2. Carbochlorination of Bauxite

#### 3.2.1. Effect of Chlorination Time

The chlorination time exhibits a significant influence on the chlorination efficiency of bauxite. A series of experiments were conducted under specific conditions to investigate this relationship, with bauxite pellets weighing 100 g, a gas flow of 10 L/min, an oxygen content of 15%, a chlorination temperature of 1000 °C, and a C/O molar ratio of 2.206. The chlorination time varied between 15 min and 75 min. The correlation between chlorination time and the bauxite chlorination efficiency is depicted in Figure 4. Additionally, the XRD patterns are presented in Figure 5.

Figure 4 demonstrates that prolonging chlorination time enhances the chlorination process of bauxite. The chlorination efficiency of Al_2_O_3_ exhibits the most rapid growth rate within the initial 30 min of the reaction, achieving an efficiency of 80.10% at 60 min. Further extension beyond this point has a little impact, indicating that the chlorination volatilization of Al_2_O_3_ can essentially be accomplished within 60 min. The chlorination efficiency of SiO_2_ increases rapidly from 38.45% at 15 min to 61.39% at 60 min. However, when extended to 75 min, the chlorination efficiency of SiO_2_ begins to decrease. This is due to the participation of CaO, MgO, and Li_2_O in bauxite in the carbochlorination process, which converts into high-boiling chlorides. CaCl_2_, MgCl_2_, and LiCl accumulate on the surface of the pellets, inhibiting the mass transfer process of chlorine inside and outside the pellets, thereby negatively affecting the carbochlorination reaction.

As illustrated in Figure 5, the diffraction peaks of diaspore in the raw material vanish, while those of corundum markedly increase. This phenomenon can be ascribed to the conversion of AlOOH to α-Al_2_O_3_ via a dehydration reaction at 500 °C. When the chlorination time is 15 min, the diffraction peaks of kaolinite disappear and are replaced by those of sillimanite and mullite. As the chlorination time extends, the diffraction peaks of mullite gradually diminish while those of cristobalite progressively increase. This suggests that the aluminum of mullite initially participates in the carbochlorination process, while the unreacted silicon remains in the cristobalite. Within 15~45 min, the diffraction peaks of corundum and quartz gradually decrease, indicating that the carbochlorination reaction of Al_2_O_3_ and free-state SiO_2_ is significant. However, with a further extension of the chlorination time, the diffraction peaks of maghemite are no longer detected, and the diffraction peak strengths of quartz and alumina are enhanced again. This is due to the decomposition of sillimanite into Al_2_O_3_ and free-state SiO_2_, suggesting that sillimanite is completely decomposed at 60 min. This is also why the growth rate of Al_2_O_3_ chlorination slows down and the chlorination efficiency of SiO_2_ decreases over an extended period of chlorination time. Based on these observations, 60 min is the optimal chlorination time for subsequent experiments.

#### 3.2.2. Effect of Coking Coal Addition

The influence of coking coal addition on the chlorination efficiency of bauxite was studied by carbochlorination at 1000 °C for 60 min with 100 g of bauxite pellets, a gas flow rate of 10 L/min, an oxygen content of 15%, and a molar ratio of C/O in the range of 1.103~2.482, as indicated in Figure 6.

The result shows that the chlorination efficiency of bauxite improves with an increase in the C/O molar ratio. Introducing an appropriate amount of coking coal into the chlorination system reduces the oxygen potential and facilitates the forward reaction. However, an excessively high C/O molar ratio, resulting from the addition of an excessive amount of coking coal to the bauxite pellets, not only fails to enhance the chlorination efficiency but also leads to waste and negatively affects the diffusion rate of gas within and outside the pellets [15]. The optimal C/O molar ratio is 2.206, under which the chlorination efficiencies of Al_2_O_3_ and SiO_2_ are recorded at 92.71% and 83.44%, respectively.

#### 3.2.3. Effect of Chlorination Temperature

A series of experiments were conducted to examine the impact of chlorination temperature on the chlorination efficiency of bauxite within a temperature range of 700~1000 °C, with 100 g of bauxite pellets, a gas flow rate of 10 L/min, an oxygen content of 15%, a C/O molar ratio of 2.206, and a chlorination time of 60 min; the results are depicted in Figure 7. The XRD patterns are given in Figure 8.

As illustrated in Figure 7, the chlorination efficiency of bauxite initially escalates significantly before stabilizing with an increase in chlorination temperature. When the temperature is raised from 700 to 1000 °C, the chlorination efficiency of Al_2_O_3_ surges rapidly to 92.71%. However, any further increase in temperature results in a slight enhancement of the chlorination efficiency of Al_2_O_3_. The chlorination efficiency of SiO_2_ exhibits a notable increase from 68.95% to 82.41% within the range of 700~900 °C. Upon continued heating to 1100 °C, the SiO_2_ chlorination efficiency experiences minor changes due to the substantial release of cristobalite after 900 °C.

The XRD spectral pattern for the chlorinated residues, recorded at varying chlorination temperatures, is presented in Figure 8. Notably, no characteristic peaks of diaspore and kaolinite were detected in the chlorinated residues at 900 °C, in contrast to those observed in the raw material. Conversely, a significant number of diffraction peaks corresponding to corundum, mullite, and cristobalite were evident. This phenomenon can be attributed to the dehydration reaction of diaspore, leading to the formation of Al_2_O_3_. At elevated temperatures, kaolinite undergoes a transformation into mullite. Since Al in mullite is more reactive than Si during the chlorination process, the residual Si components predominantly exist in the cristobalite phase. The intensity of the diffraction peaks for corundum and quartz increased between 700 °C and 800 °C but diminished within the temperature range of 900 to 1100 °C. This suggests that free-state Al_2_O_3_ and SiO_2_ dominate in the carbochlorination process up to 900 °C. However, when temperatures exceeded 900 °C, sillimanite decomposed into corundum and quartz, while mullite transformed into corundum and cristobalite. These changes led to a decelerated rate of the carbochlorination reaction involving Al_2_O_3_ and SiO_2_, aligning with their respective chlorination efficiency trends depicted in Figure 7. Consequently, a temperature of 1000 °C was determined to be optimal for subsequent reactions.

#### 3.2.4. Effect of Gas Flow

Under the conditions of 100 g bauxite pellets, a chlorination temperature of 1000 °C, a chlorination time of 60 min, an oxygen content of 15%, and a C/O molar ratio of 2.206, the effects of various gas flow were studied ranging from 8 L/min to 16 L/min, as shown in Figure 9. It was observed that the chlorination efficiency of Al_2_O_3_ and SiO_2_ in bauxite improved as the gas flow rate increased from 8 L/min to 14 L/min. Nevertheless, further increases in the gas flow resulted in a decline in chlorination efficiency. This trend is likely due to the enhancement of gas flow. Unfortunately, the formation of forced convectionreduces the gas’s residence time in the furnace, which is not favorable for the carbochlorination process. As a consequence, the optimal gas flow rate is 14 L/min and the chlorination efficiencies of Al_2_O_3_ and SiO_2_ are 94.93% and 86.32%, respectively.

#### 3.2.5. Effect of Oxygen Content

Experiments to investigate the oxygen content effect were carried out in an oxygen content range of 0~20% under the conditions of 100 g of bauxite pellets, a chlorination temperature of 1000 °C, a chlorination time of 60 min, a gas flow of 14 L/min, and a molar ratio of C/O of 2.206; the results are shown in Figure 10. The chlorination efficiencies of Al_2_O_3_ and SiO_2_ witnessed an increase from 76.08% and 61.18% (at an oxygen content of 0%) to 94.93% and 86.32% (at an oxygen content of 15%). However, it marginally reduced the oxygen content by 20%. The appropriate introduction of oxygen supplied heat to the carbochlorination system, thus promoting the carbochlorination process of bauxite. Conversely, excess oxygen consumed part of the carbon, leading to an insufficient carbon content that was not conducive to the reaction. In consequence, the chlorination efficiencies of Al_2_O_3_ and SiO_2_ reached the maximum value of 94.93% and 86.32%, respectively.

### 3.3. Carbochlorination Kinetics of Low-Grade Bauxite

#### 3.3.1. Effects of Chlorination Temperature and Time

The relationship between chlorination temperature and time on the chlorination efficiency of Al_2_O_3_ and SiO_2_ in bauxite at temperatures of 800~1100 °C is shown in Figure 11. The results show that the chlorination efficiency of Al_2_O_3_ grew obviously during the first 15 min. At a temperature of 800 °C, the chlorination efficiency increased moderately from 67.76% at 30 min to 77.58% at 60 min. When increasing the temperature to 1100 °C, the chlorination efficiency of Al_2_O_3_ increased to 89.08% at 30 min, a value significantly higher than that achieved through calcination at 800 °C for 120 min. Similarly, the chlorination efficiency of SiO_2_ followed a consistent pattern. The chlorination efficiency of SiO_2_ only increased from 57.74% at 30 min to 72.43% at 60 min. Nevertheless, raising the temperature to 1100 °C led to a dramatic increase to 78.66% at 30 min. This clearly demonstrates that an increase in temperature can effectively enhance the carbochlorination process of Al_2_O_3_ and SiO_2_.

#### 3.3.2. Effects of Oxygen Content and Chlorination Time

The correlation between chlorination temperature and time regarding the chlorination efficiency of Al_2_O_3_ and SiO_2_ in bauxite at an oxygen content of 0~20% is depicted in Figure 12. Within the first 15 min, the chlorination efficiency of Al_2_O_3_ and SiO_2_ increased rapidly, but with the further extension of time, the growth rate of chlorination efficiency gradually slowed. Without oxygen, the chlorination efficiency of Al_2_O_3_ and SiO_2_ increased from 65.59% and 51.52% at 30 min to 76.08% and 61.18% at 60 min. In contrast, when the oxygen content was increased to 15%, these values significantly improved to 86.30% and 74.51%. This enhancement can be ascribed to the introduction of oxygen, which provides heat to the carbochlorination system, thereby further enhancing the carbochlorination process of bauxite.

#### 3.3.3. Determination of Chlorination Kinetic Model

Based on the hypothesis that bauxite pellets are dense solid particles with identical surface chemical activity, the carbochlorination reactions of bauxite are described as follows:(1)Cl_2_ permeated the surface of bauxite pellets via the gas phase diffusion boundary layer.(2)Cl_2_ was diffused through porous solid media and subsequently adsorbed by the aluminum–silicon phase in the pellets.(3)The carbochlorination reaction between Cl_2_ and the aluminum–silicon phase resulted in the production of gaseous AlCl_3_ and SiCl_4_.(4)AlCl_3_ and SiCl_4_ diffused into the gas phase through the porous medium layer.(5)AlCl_3_ and SiCl_4_ underwent diffusion from the gas boundary layer into the surrounding air.

In this study, carbochlorination kinetic models at varying temperatures were discussed. Furthermore, given that the introduction of oxygen altered the gas phase composition of the chlorination system, carbochlorination kinetics models at various oxygen contents were subsequently investigated.

The shrinkage of bauxite pellets during carbochlorination can be effectively modeled using an unreacted shrinking core approach to elucidate the reaction and compute the carbochlorination kinetics. The rate-determining steps involved in gas–solid reactions probably include a chemical reaction, external diffusion, or internal diffusion. Consequently, the kinetic equations can be articulated as follows:(1)Kinetics equation of chemical reaction control:
(4)$$1-{\left(1-x\right)}^{1/3}=k_{1}t$$

(2)Kinetics equation of external diffusion control:


(5)
$$1-{\left(1-x\right)}^{2/3}=k_{2}t$$


(3)Kinetics equation of internal diffusion control:

(6)$$1+2\left(1-x\right)-3{\left(1-x\right)}^{2/3}=k_{3}t$$
where *k*_1_, *k*_2_, and *k*_3_ are apparent reaction rate constants under different rate-determining steps, and *x* is the chlorination efficiency of Al_2_O_3_ or SiO_2_. Kinetic functions of Al_2_O_3_ and SiO_2_ can be determined on the basis of linear fitting with chlorination data.

The apparent activation energy of the bauxite carbochlorination reaction was calculated by the Arrhenius equation:(7)$$\ln k=-\frac{Ea}{RT}+\ln A$$
where *k* represents the apparent reaction rate constants; *E*a represents the apparent activation energy, kJ·mol^−1^; *R* represents the ideal gas constant, kJ·mol^−1^·K^−1^; *T* represents the absolute temperature, K; and *A* represents the pre-exponential factor, min^−1^.

Moreover, the apparent reaction order of bauxite carbochlorination reaction was calculated by van’t Hoff differentiation:(8)$$\ln k=n\ln C_{\left(O_{2}\right)}+\ln Z$$
where *k* represents the apparent reaction rate constants; n is the apparent reaction order; *C*_(O2)_ is the oxygen concentration represented by oxygen partial pressure; and *Z* is a constant, kJ·mol^−1^.

Drawing from the above research findings, it can be inferred that the chlorination rate of Al_2_O_3_ and SiO_2_ is potentially modulated by the diffusion of gasses through the product layer. Consequently, Equation (6) was selected to fit the carbochlorination data of Al_2_O_3_ and SiO_2_ at various temperatures (Figure 11), as depicted in Figure 13.

Figure 13a,b show the intrinsic kinetics of Al_2_O_3_ and SiO_2_ carbochlorination at various temperatures. The Arrhenius plots are plotted according to the rate constant k obtained by the linear fitting slope, as presented in Figure 13c,d. It can be clearly seen that the carbochlorination reaction of Al_2_O_3_ and SiO_2_ in bauxite can be divided into two stages with apparent differences in chlorination efficiency, with 15~30 min as the cut-off point. Upon calculation, it has been ascertained that in the temperature range of 800~1100 °C, the apparent activation energy of Al_2_O_3_ chlorination is 34.281 kJ/mol at the first stage and 28.022 kJ/mol at the second stage. The apparent activation energy of SiO_2_ chlorination is 41.343 kJ/mol at the first stage and 13.095 kJ/mol at the second stage. The pre-exponential factor A can be ascertained from the intercept of the fitted lines in Figure 13c,d. Consequently, the kinetic equations for the Al_2_O_3_ carbochlorination reaction during the low-grade bauxite carbochlorination can be articulated as follows:

0~15 min:(9)$$1+2\left(1-x\right)-3{\left(1-x\right)}^{2/3}=0.599e^{\frac{-34281}{RT}}t$$

30~60 min:(10)$$1+2\left(1-x\right)-3{\left(1-x\right)}^{2/3}=0.0789e^{\frac{-28022}{RT}}t$$

The kinetic equations for the SiO_2_ carbochlorination reaction during the low-grade bauxite carbochlorination can be articulated as follows:

0~15 min:(11)$$1+2\left(1-x\right)-3{\left(1-x\right)}^{2/3}=0.693e^{\frac{-41343}{RT}}t$$

30~60 min:(12)$$1+2\left(1-x\right)-3{\left(1-x\right)}^{2/3}=0.0187e^{\frac{-13095}{RT}}t$$

The chlorination data for Al_2_O_3_ and SiO_2_ in bauxite, under varying oxygen contents (Figure 12), were fitted into Equation (6) as depicted in Figure 14. The findings suggest that fitting the chlorination rates of Al_2_O_3_ and SiO_2_ is optimal when the oxygen content ranges from 0 to 15%. This implies that internal diffusion control is the rate-determining step for Al_2_O_3_ and SiO_2_ carbochlorination.

Utilizing the internal diffusion model, the intrinsic kinetics of Al_2_O_3_ and SiO_2_ carbochlorination reactions under various oxygen contents were plotted, as depicted in Figure 14a,b. The relationship between ln*k* and ln[*p*(O_2_)/*p*^0^] of Al_2_O_3_ and SiO_2_ is further drawn in Figure 14c,d, obtaining the apparent reaction order of the Al_2_O_3_ and SiO_2_ carbochlorination reaction from the slope of the linear fit. The apparent reaction orders of Al_2_O_3_ at the first and second stages are 0.58 and 0.16, respectively. The apparent reaction orders of SiO_2_ at the first and second stages are 0.62 and 0.52, respectively. These findings indicate that the influence of oxygen content on Al_2_O_3_ chlorination efficiency surpasses that of SiO_2_.

### 3.4. Carbochlorination Mechanism of Low-Grade Bauxite

The results of carbochlorination experiments, coupled with the characterization of chlorinated residues, reveal that a variety of mineral phases participate in bauxite carbochlorination. A deeper understanding of the mechanism underlying the carbochlorination of low-grade bauxite is crucial for informing industrial applications of this process.

Figure 15 illustrates that the diaspore decomposes into corundum under heat during bauxite carbochlorination. The preferential participation of certain Al components in the reaction converts kaolinite into mullite and sillimanite, with any remaining Si existing as a cristobalite phase. As carbochlorination progresses, mullite decomposes into corundum and cristobalite, while sillimanite decomposes into corundum and quartz. Ultimately, corundum is converted into AlCl_3_, and quartz and cristobalite are transformed into SiCl_4_. Bauxite contains minor impurities such as Fe_2_O_3_, CaO, and MgO, converted into high-boiling-point compounds FeCl_3_, CaCl_2_, and MgCl_2_. These compounds cannot volatilize and, therefore, remain solids in the chlorinated residue, which the washing process can remove.

## 4. Conclusions

(1)The feasibility of recovering aluminum and silicon from low-grade bauxite via carbochlorination was verified through thermodynamic analysis. The carbochlorination of aluminum has precedence over that of silicon.(2)The primary determinants influencing the carbochlorination process of low-grade bauxite include the chlorination time, amount of added coking coal, chlorination temperature, gas flow, and oxygen content. Optimal results are achieved when operating under conditions of a chlorination temperature of 1000 °C, a chlorination time of 60 min, a gas flow of 14 L/min, a C/O molar ratio of 2.206, and an oxygen content of 15%, with which the chlorination efficiency for Al_2_O_3_ and SiO_2_ can reach 94.93% and 86.32%, respectively.(3)The kinetic analysis of the carbochlorination process involving Al_2_O_3_ and SiO_2_ in bauxite reveals that the reaction adheres to a shrinking, unreacted core model. This model is primarily governed by gas diffusion within the product layer, encompassing two distinct stages. The apparent activation energies for the initial and subsequent stages of the Al_2_O_3_ chlorination reaction, occurring at temperatures ranging from 800 to 1100 °C, are 34.281 kJ/mol and 28.022 kJ/mol, respectively. Likewise, the apparent activation energies for the two stages of the SiO_2_ chlorination reaction stand at 41.343 kJ/mol and 13.095 kJ/mol, respectively.(4)The observed variation in oxygen content, ranging from 0% to 15%, yielded apparent reaction orders of 0.58 and 0.16 for the first and second stages of Al_2_O_3_ chlorination, respectively. Similarly, for SiO_2_ chlorination, the evident reaction orders were determined to be 0.62 and 0.52 for the two stages, respectively.

## Figures and Tables

**Figure 1 materials-17-03613-f001:**
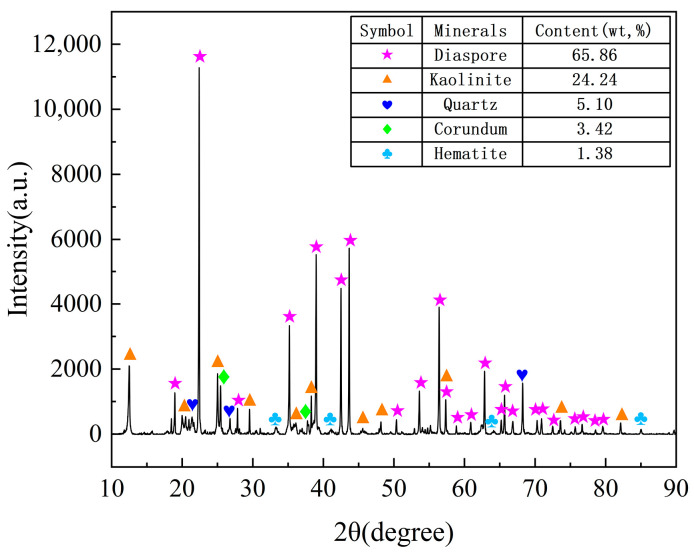
XRD pattern of low-grade bauxite and of mineral contents.

**Figure 2 materials-17-03613-f002:**
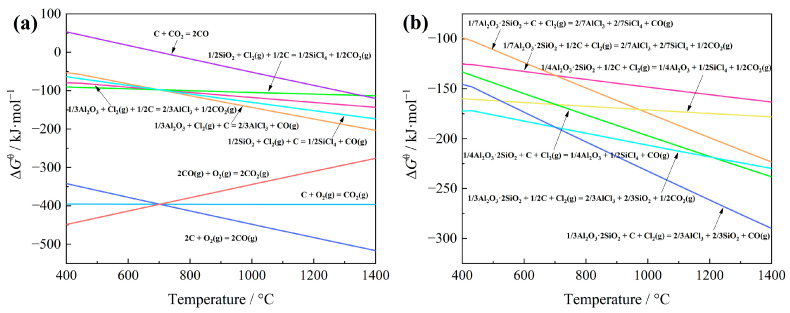
Gibbs free energy change (standard) ∆*G*^θ^ for carbochlorination reactions of Al, Si compounds in bauxite at different temperatures: (**a**) the reactions of Al_2_O_3_ and SiO_2_; (**b**) the reactions of Al_2_O_3_·2SiO_2_.

**Figure 3 materials-17-03613-f003:**
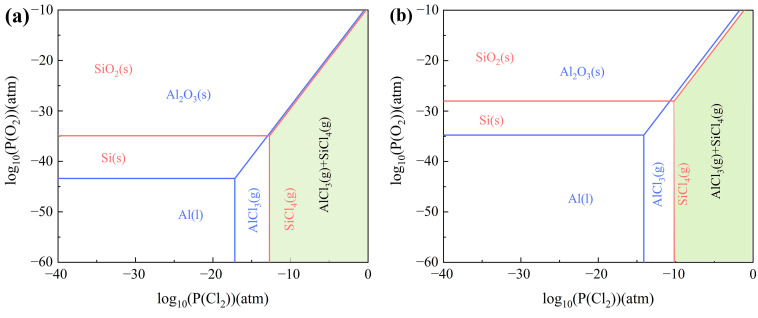
Predominance diagram of Al-Si-O-Cl system at (**a**) 800 °C; (**b**) 1000 °C.

**Figure 4 materials-17-03613-f004:**
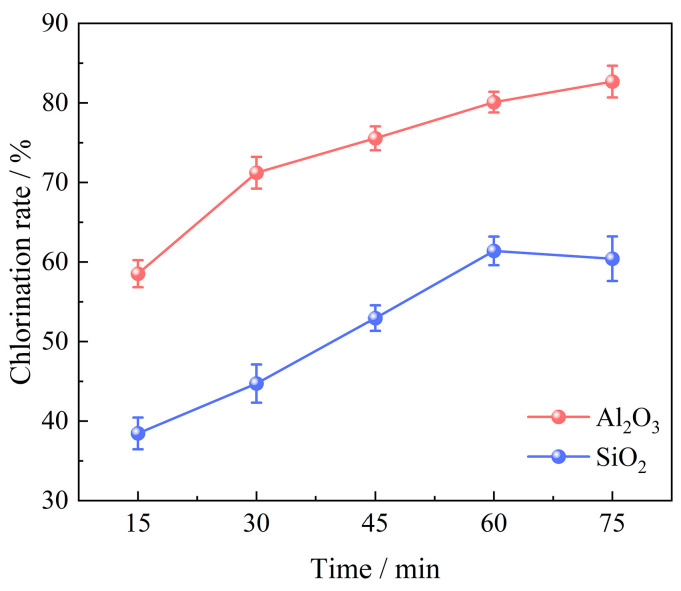
Effect of chlorination time on chlorination efficiency of bauxite.

**Figure 5 materials-17-03613-f005:**
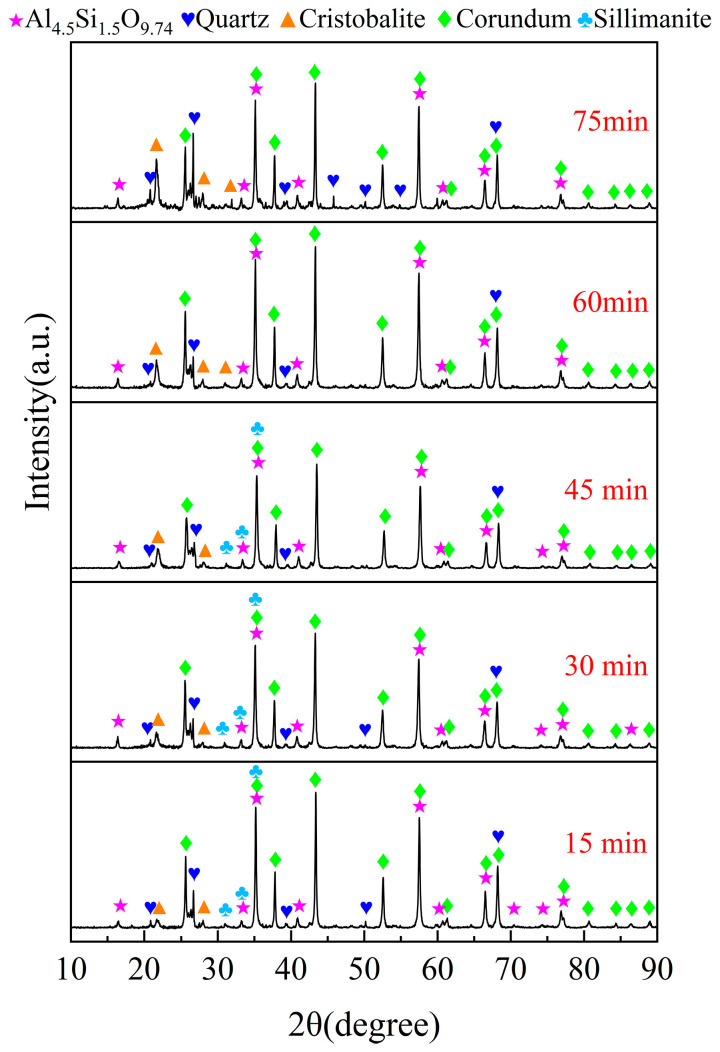
XRD patterns of the chlorinated residues at different chlorination times.

**Figure 6 materials-17-03613-f006:**
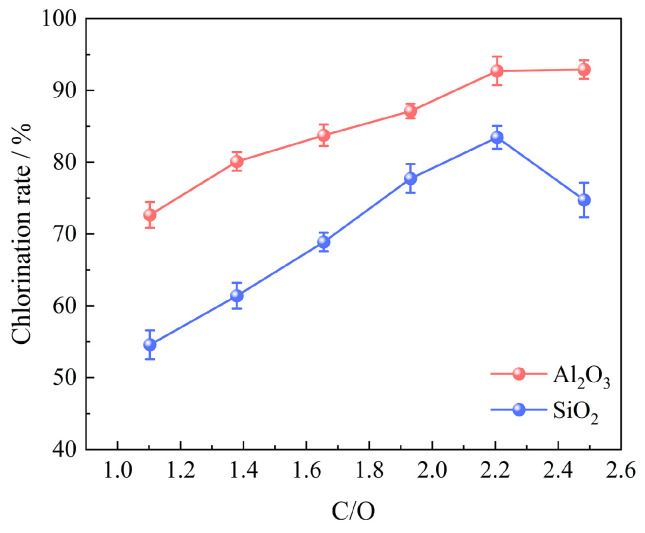
Effect of coking coal addition on chlorination efficiency of bauxite.

**Figure 7 materials-17-03613-f007:**
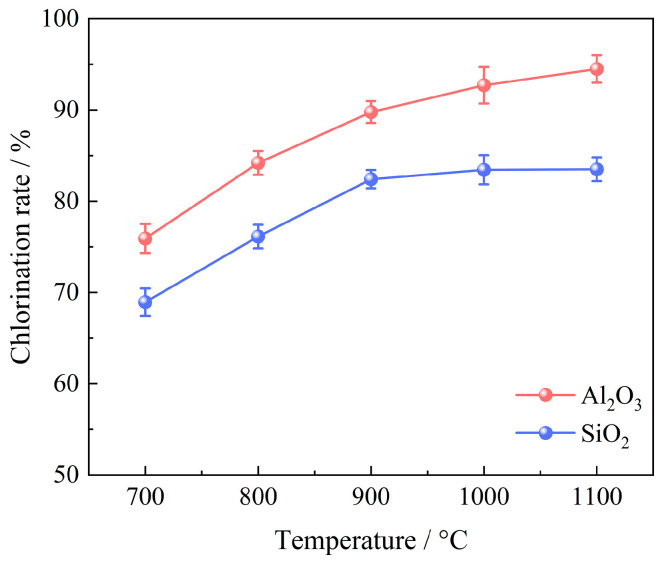
Effect of chlorination temperature on chlorination efficiency of bauxite.

**Figure 8 materials-17-03613-f008:**
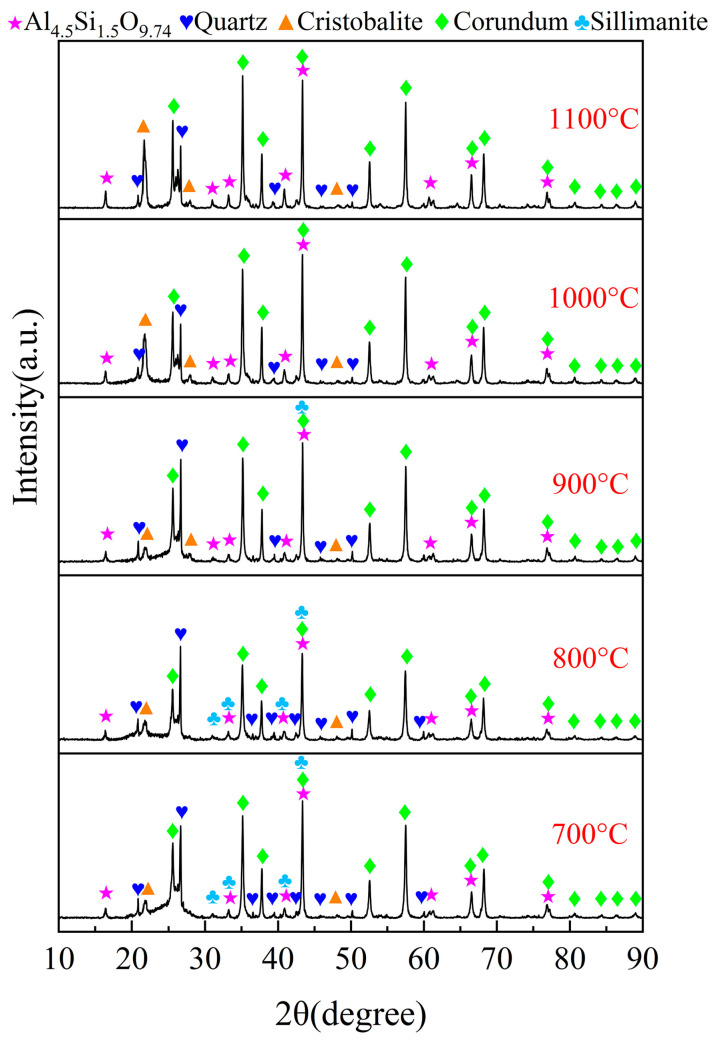
XRD patterns of the chlorinated residues at different chlorination temperature.

**Figure 9 materials-17-03613-f009:**
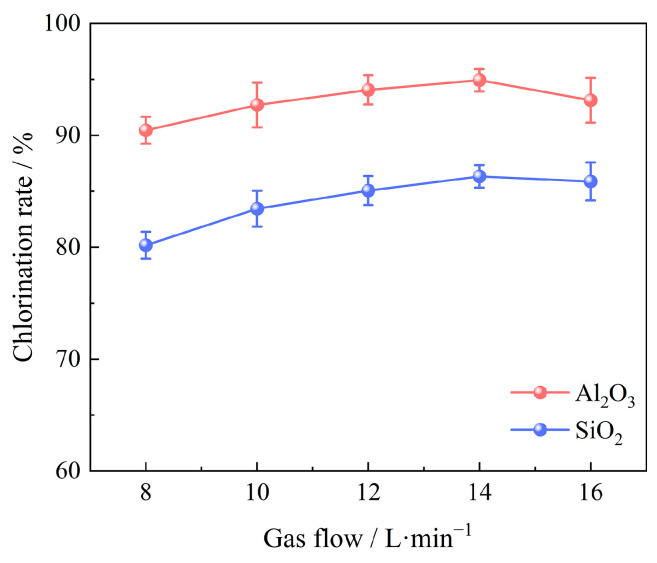
Effect of gas flow on chlorination efficiency of bauxite.

**Figure 10 materials-17-03613-f010:**
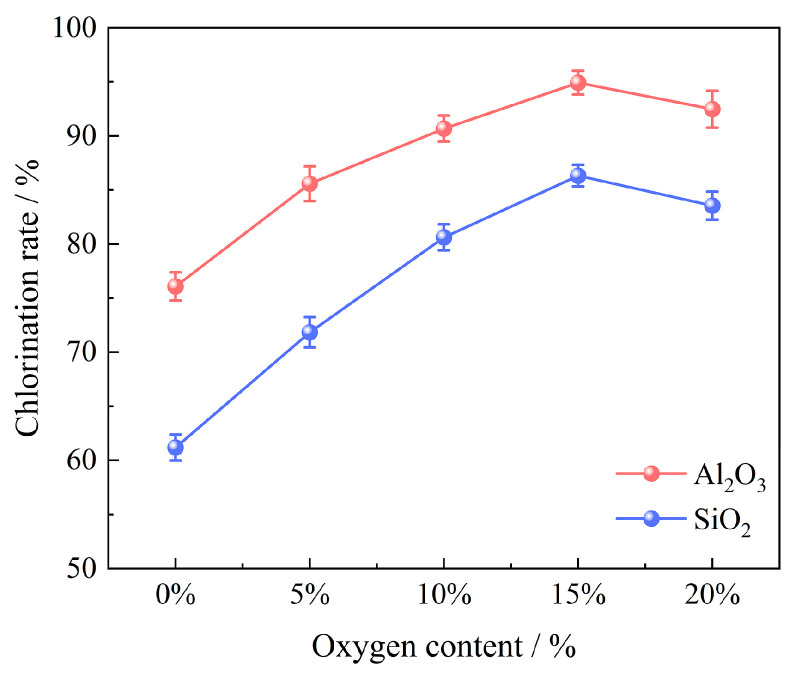
Effect of oxygen content on chlorination efficiency of bauxite.

**Figure 11 materials-17-03613-f011:**
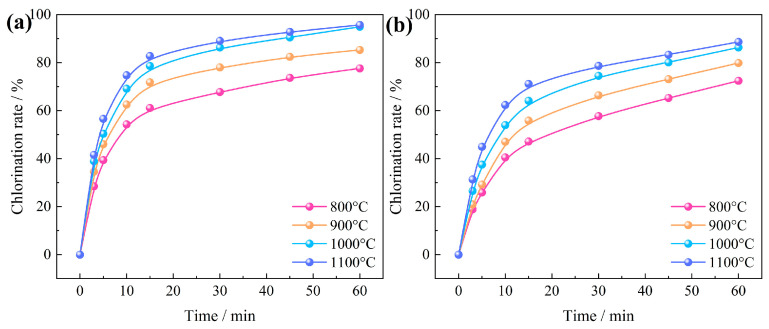
Relationship between chlorination temperature and time on the chlorination efficiency of bauxite: (**a**) Al_2_O_3_; (**b**) SiO_2_.

**Figure 12 materials-17-03613-f012:**
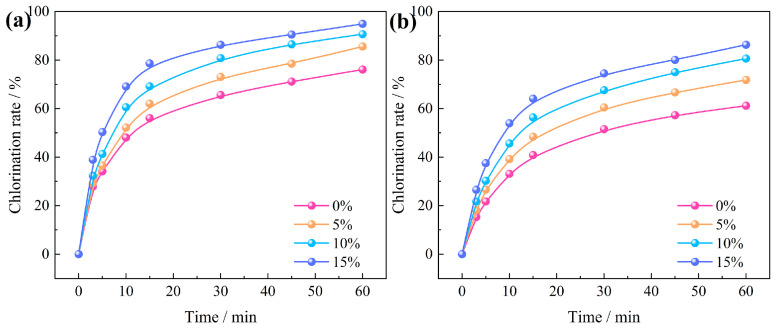
Relationship between oxygen content and chlorination time on the chlorination efficiency of bauxite: (**a**) Al_2_O_3_; (**b**) SiO_2_.

**Figure 13 materials-17-03613-f013:**
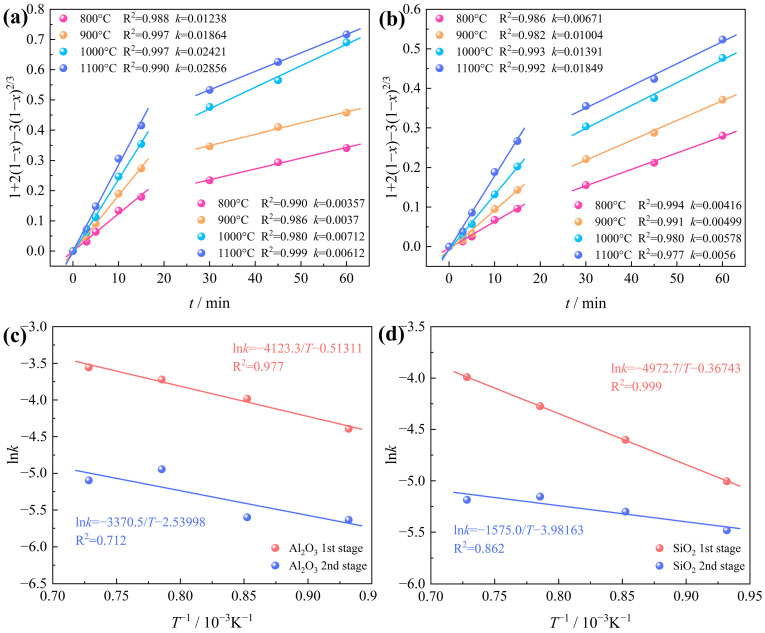
Intrinsic kinetics of bauxite carbochlorination at various temperatures: (**a**) Al_2_O_3_; (**b**) SiO_2_; Arrhenius plots of the rate constants of (**c**) Al_2_O_3_; (**d**) SiO_2_.

**Figure 14 materials-17-03613-f014:**
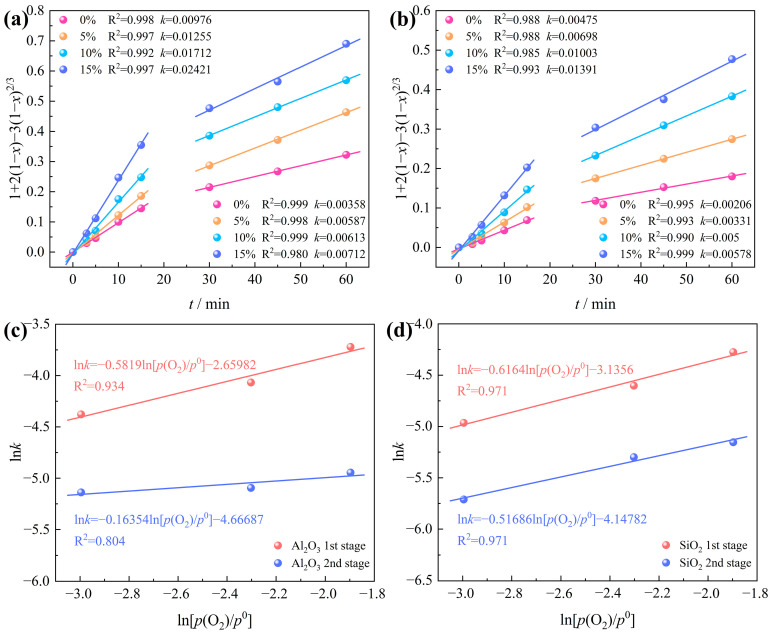
Intrinsic kinetics of bauxite carbochlorination at various oxygen contents: (**a**) Al_2_O_3_; (**b**) SiO_2_; plots of ln*k* versus ln[*p*(O_2_)/*p*^0^] of (**c**) Al_2_O_3_; (**d**) SiO_2_.

**Figure 15 materials-17-03613-f015:**
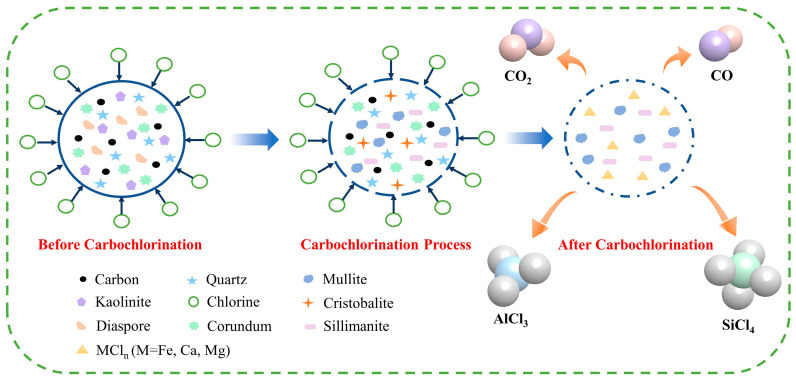
Schematic diagram of low-grade bauxite carbochlorination.

**Table 1 materials-17-03613-t001:** Chemical compositions of bauxite.

Component	Al_2_O_3_	SiO_2_	CaO	Fe_2_O_3_	TiO_2_	MgO	K_2_O	A/S
Content, wt.%	61.11	22.49	1.25	8.42	3.73	0.37	1.08	2.72

## Data Availability

The original contributions presented in the study are included in the article; further inquiries can be directed to the corresponding authors.

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
