# Peer review of "Co-Extraction of Aluminum and Silicon and Kinetics Analysis in Carbochlorination Process of Low-Grade Bauxite"

_materials, 2024, doi:10.3390/ma17143613_

Round 1
Reviewer 1 Report
Comments and Suggestions for Authors
The work of Zhao et al. investigates a carbochlorination process of low-grade bauxite ore. The study investigates with a systematic approach the effect that various parameters can have on the reactions involved in the process. The manuscript seems best suited for other MDPI journals such as “minerals”. However, in my opinion it can still be published if major revisions are implemented. Some suggestions include the following:
Table 1: are these results from ICP-MS complete digestion or from XRF? What was the protocol used to obtain those data?
Figure 1 would benefit from a quantification of the minerals, not only of the qualitative assessment
Section 2.2. lack details such as what was the coal used, the starch etc. Were these pellets granulated with water?
In Equation 1, how was “w”, i.e. the chlorinated fraction, determined? If a Rietveld refinement was performed then the details need to be provided. If not, then it needs to be explained how it was determined.
I am not sure of what is the relevance of the list of all the reactions in Eq.4 to 20. These could go in an appendix and the authors should instead explain what are the underlying assumptions that went into the modelling with Factsage
Section 3 is quite long and would benefit the reader and the message if it could be somewhat shortened or made more concise. Table 2 and Table 4 can be eliminated if they report the same data of Figure 13 and 14.
References. They need to be completely revised. The view of the literature is limited, with more than 65% citations from Chinese authorship. Some references are not available to international readership and/or are not relevant such as Ref. 4, 5, 8, 12, 13, 15, 19, 24 etc.
Reviewer 2 Report
Comments and Suggestions for Authors
I have the following comments:
Chapter 21. Materials:
1. How many samples were taken for analysis?
2. No description of the methodology XRF, ICP-MS, and in the case of the XRD method, it was only mentioned in the figure caption.
3. No explanation why two methods were used to determine chemical composition.
4. The sum of the components in table 1 is 101.2652 wt. %.
5. Why are the values ​​given with different accuracies in table 1?
Comments on the Quality of English Language
Minor editing of English language required.
Reviewer 3 Report
Comments and Suggestions for Authors
This paper describes the co-extraction of aluminum and silicon from low-grade bauxite by the carbochlorination process. The combination of the detailed kinetic analysis and the careful experimentation provides the deep insights into the carbochlorination process.
The reviewer recommends Accept for this paper after some minor corrections for simple typological errors shown below.
(1) Subscript "4" for "SiCl4" (P.5, L.163).
(2) Put a space between "Figure" and "7" (P.8, L.236)
In addition, the reviewer is interested in the fact that the chlorination efficiency is rather strongly influenced by conditions such as C/O ratio (Figure 6) and oxygen content (Figure 10). If possible, add comments to the discussion, based on the findings of this study, on how the optimum conditions change when the grade of the starting bauxite, i.e. Al/Si ratio, is changed.
Round 2
Reviewer 1 Report
Comments and Suggestions for Authors
See comments on references
Comments on the Quality of English LanguageEnglish is good
Author Response
Comments 1: See comments on references.
Response 1: Thank you for your valuable suggestion. All the references you mentioned previously have been deleted as per request. Additionally, we have reviewed and amended other references to ensure their accuracy and relevance. The revised references are as follows.
References
- Shah, S. S.; Palmieri, M. C.; Sponchiado, S. R. P.; Bevilaqua, D., A sustainable approach on biomining of low-grade bauxite by P. simplicissimum using molasses medium. Brazilian Journal of Microbiology 2022, 53, (2), 831-843.
- Vakilchap, F.; Mousavi, S. M.; Shojaosadati, S. A., Role of Aspergillus niger in recovery enhancement of valuable metals from produced red mud in Bayer process. Bioresource Technology 2016, 218, 991-998.
- Yang, Q. C.; Zhang, F.; Deng, X. J.; Guo, H. C.; Zhang, C.; Shi, C. S.; Zeng, M., Extraction of alumina from alumina rich coal gangue by a hydro-chemical process. Royal Society Open Science 2020, 7, (4).
- Xu, Y. P.; Chen, C. Y.; Lan, Y. P.; Wang, L. Z.; Li, J. Q., Desilication and recycling of alkali-silicate solution seeded with red mud for low-grade bauxite utilization. Journal of Materials Research and Technology 2020, 9, (4), 7418-7426.
- Birinci, M.; Gök, R., Characterization and flotation of low-grade boehmitic bauxite ore from Seydisehir (Konya, Turkey). Minerals Engineering 2021, 161.
- Lakshmanan, V. I.; Sridhar, R.; Chen, J.; Halim, M. A., Development of Mixed-Chloride Hydrometallurgical Processes for the Recovery of Value Metals from Various Resources. Transactions of the Indian Institute of Metals 2016, 69, (1), 39-50.
- Grjotheim, K., Aluminium electrolysis: The chemistry of the Hall-Hult process. 1977.
- Liang, X. M., Development of Alcoa's aluminum electrolysis technology (in Chinese). Light Metals 2020, (1), 1-10.
- Wyndham, R. Controlled carbo-chlorination of kaolinitic ores. US4083927-A; ZA7802097-A; FR2422723-A; CA1105712-A; GB1601085-A, 1978.
- Reynolds, J. E.; Williams, A. R. Process for chlorinating clay and bauxite. US4288414-A, 1981.
- Wegner, G.; Blumenthal, G. Thermal chlorination of raw material contg. aluminium - as silicate and/or oxide with gaseous silicon tetra:chloride after chemical activation. DD212246-A, 1984.
- Wang, L.; Zhao, X. X.; Zhang, Z. M.; Zhang, T. A.; Lv, G. Z.; Dou, Z. H.; Zhao, A. C.; Zhang, X. Y., Synergistic separation and extraction of valuable elements from high-alumina fly ash with carbochlorination method. Transactions of Nonferrous Metals Society of China 2023, 1-20.
- Bale, C. W.; Bélisle, E.; Chartrand, P.; Decterov, S. A.; Eriksson, G.; Gheribi, A. E.; Hack, K.; Jung, I. H.; Kang, Y. B.; Melançon, J.; Pelton, A. D.; Petersen, S.; Robelin, C.; Sangster, J.; Spencer, P.; Van Ende, M. A., FactSage thermochemical software and databases, 2010-2016. Calphad-Computer Coupling of Phase Diagrams and Thermochemistry 2016, 54, 35-53.
- Lahijani, P.; Zainal, Z. A.; Mohammadi, M.; Mohamed, A. R., Conversion of the greenhouse gas CO2 to the fuel gas CO via the Boudouard reaction: A review. Renewable & Sustainable Energy Reviews 2015, 41, 615-632.
- Deng, P.; Li, L.; Jia, Y. Q.; Liu, D. C.; Jiang, W. L.; Kong, L. X., Chlorination behavior of low-grade titanium slag in AlCl3-NaCl molten salt. Jom 2022, 74, (1), 213-221.
- Zhao, X. X.; Wang, L.; Cheng, T. H.; Liu, Y.; Zhang, T. G.; Zhao, Q. Y., Synergistic extraction of valuable elements from high-alumina fly ash via carbochlorination. Journal of Sustainable Metallurgy 2024.

Reviewer 2 Report
Comments and Suggestions for Authors
All my comments were taken into account.
Author Response
Comments 1: All my comments were taken into account.
Response 1: Thank you for your valuable advice. I look forward to the eventual publication of this manuscript in Materials with your help.